# A qualitative study comparing experiences of the surgical safety checklist in hospitals in high-income and low-income countries

Emma-Louise Aveling,[1] Peter McCulloch,[2] Mary Dixon-Woods[1]

## ABSTRACT

**Objective:** Bold claims have been made for the ability of the WHO surgical checklist to reduce surgical morbidity and mortality and improve patient safety regardless of the setting. Little is known about how far the challenges faced by low-income countries are the same as those in high-income countries or different. We aimed to identify and compare the influences on checklist implementation and compliance in the UK and Africa.

**Design:** Ethnographic study involving observations, interviews and collection of documents. Thematic analysis of the data.

**Setting:** Operating theatres in one African university hospital and two UK university hospitals.

**Participants:** 112 h of observations were undertaken. Interviews with 39 theatre and administrative staff were conducted.

**Results:** Many staff saw value in the checklist in the UK and African hospitals. Some resentment was present in all settings, linked to conflicts between the philosophy behind the checklist and the realities of local cultural, social and economic contexts. Compliance—involving use, completeness and fidelity—was considerably higher, though not perfect, in the UK settings. In these hospitals, compliance was supported by established structures and systems, and was not significantly undermined by major resource constraints; the same was not true of the low-income context. Hierarchical relationships were a major barrier to implementation in all settings, but were more marked in the low-income setting. Introducing a checklist in a professional environment characterised by a lack of accountability and transparency could make the staff feel jeopardised legally, professionally, and personally, and it encouraged them to make misleading records of what had actually been done.

**Conclusions:** Surgical checklist implementation is likely to be optimised, regardless of the setting, when used as a tool in multifaceted cultural and organisational programmes to strengthen patient safety. It cannot be assumed that the introduction of a checklist will automatically lead to improved communication and clinical processes.

[1]Department of Health Sciences, University of Leicester, Leicester, UK
[2]Nuffield Department of Surgical Science, University of Oxford, Oxford, UK

**Correspondence to**
Dr Emma-Louise Aveling; eea5@le.ac.uk

## ARTICLE SUMMARY

### Article focus
- Bold claims have been made for the ability of the WHO surgical checklist to reduce surgical morbidity and mortality and improve patient safety.
- We aimed to identify and compare influences on checklist implementation and compliance in operating theatres in two UK hospitals and one African hospital.

### Key messages
- Consistent use, completion and fidelity of checklist deployment are not straightforward in any setting, but may be higher in the two UK hospitals compared with the hospital in a low-income country.
- Contrary to claims in early studies of the checklist, additional resources and changes to clinical systems may be needed to secure compliance with the checklist and its principles in low-income contexts.
- The checklist is no magic bullet; improvements in communication and the quality of team interactions do not automatically follow its introduction in any setting, and the checklist may indeed introduce new, unintended risks in low-income settings.

### Strengths and limitations of this study
- Detailed, first-hand observations of the implementation of the surgical checklist in diverse settings; identification of key lessons of practical value in securing the benefits of the checklist and avoiding unintended consequences, especially in low-income countries.
- Data on outcomes were unavailable; observations conducted at different stages of implementation; some evidence of observer effect; limited number of sites; details of training were unreliable.

single episode, and will remain a key therapeutic strategy in all countries. Surgery is, unfortunately, also a major source of avoidable morbidity and mortality worldwide,[1] [2] though substantial improvements can be achieved by reducing variation in the reliability of surgical care processes.[3] Checklists are increasingly being promoted as a way to deliver such reliability, following the pilot study of the WHO's

## INTRODUCTION

Surgery provides immediate, transformative treatment for many conditions, usually in a

Surgical Safety Checklist (figure 1) which reported reductions in mortality and complications in surgical patients.[4] The checklist involves 19 separate items to be checked at three distinct points: the Sign In, before induction of anaesthesia; the Time Out, before skin incision; and the Sign Out, at the end of the procedure.

One important and unusual feature of the WHO checklist is its claim to universality. The checklist pilot study[4] was distinctive for its being conducted in high, middle and lower income settings, including one African hospital. It concluded that the checklist programme could 'improve the safety of surgical patients in diverse clinical and economic environments' (p.496). It also suggested that implementation was neither costly nor lengthy, proposing that only two items—pulse oximetry and prophylactic antibiotics—would involve commitment of significant resources; both were reported to be available at all sites in the study (including the low-income country hospitals), though used inconsistently.

A second important feature of the WHO checklist is that it combines checks for technical items (such as administration of antibiotics and use of pulse oximeters) with other so-called non-technical items (such as team introductions and confirmations of procedures) whose principal purpose is to promote aspects of teamwork, communication and situational awareness. Inclusion of the non-technical items in the checklist was influenced by research demonstrating an association between team practices (eg, communication behaviours) and improved safety processes and attitudes.[5] [6] Performing the checklist is also envisioned as a non-technical intervention in its own right: the checks are to be performed orally, and are, according to the checklist pilot study, 'intentionally designed to create a collective awareness among surgical teams of whether safety processes are being completed' (p.497).[4]

The potential for surgical safety checklists to improve safety and outcomes and generate substantial cost savings[7] has attracted global interest. In part, owing to the WHO's Safe Surgery Saves Lives campaign, around 1800 institutions are now reported to be using the checklist worldwide.[8] Studies in high-income and low-income settings have continued to report positive impacts on surgical outcomes, but the size of the reported effects varies across and within settings, possibly because of the variability in compliance.[9–13] Compliance can be heuristically distinguished into three dimensions: use, completeness and fidelity. Use refers to whether the checklist is used at all; completeness refers to the extent to which it is completed in full, without items being skipped; and fidelity refers to the extent to which items are performed as intended, with items ticked as complete only when checks have genuinely been made, at the right time and in communication with the whole team.

The checklist pilot study[4] and subsequent studies of resource-poor settings[9] were focused on assessment of outcomes, and did not characterise the implementation processes and characteristics of contexts likely to optimise checklist compliance. Yet implementing checklists, securing compliance and replicating positive outcomes necessarily require an appreciation of the sociocultural context.[14] Healthcare practitioners do not blindly follow procedures; rather, compliance often depends on workers' perceptions of effectiveness, relevance and rationale, their efforts to juggle competing professional, moral and social norms and demands and the institutional and socioeconomic contexts in which they work.[15] [16] Studies of checklist implementation in high-income countries suggest that challenges to compliance are linked to organisational safety culture,[14] and may include hierarchical structures, dismissive attitudes, absence of key team members and hurrying through checks.[17–19] Yet study of the checklist in low-income countries has been largely neglected. This is an important omission: complications and mortality following surgery in sub-Saharan Africa remain far higher than in developed countries[1] and almost 200 registered institutions in Africa are now reported to be using the checklist.[20] The specific challenges likely to be faced in such settings remain under-researched. We aimed to identify and compare influences on checklist implementation and compliance in operating theatres in hospitals in high-income and low-income countries.

## METHODS

Our study design involved ethnographic case studies[21] in three hospitals: two in the UK (anonymised as Amfield and Tolgrave) and one (anonymised as Mbile) in a low-income sub-Saharan African country. The UK centres were two large university hospitals in major urban settings that were participating in a larger research project on culture and behaviour relating to patient safety. The African centre was a university hospital serving a mixed urban and rural population of approximately 4–5 million. All three hospitals had adopted, as official policy, a checklist based on the WHO template. The checklist had been mandated for use in all UK hospitals by the National Patient Safety Agency 2 years before our study; in Mbile, it was introduced 1 year previously as part of an international collaboration to improve surgical safety. Given the sensitivity of some findings, we do not provide further details that could make participating sites or individual staff identifiable

Data collected across all three sites comprised: (1) semi-structured interviews with anaesthetists, surgeons, theatre staff, management and administrative staff and covered experiences of checklist implementation, institutional context and management of patient safety. Interviews were conducted with informed consent from participants and were recorded, translated (where required) and transcribed; (2) non-standardised observations and informal discussions with staff in operating rooms were recorded as field-notes in notebooks and later elaborated more fully and (3) collation of relevant documentation. Observations were conducted in all four operating rooms in the African site. In the UK sites, observations were undertaken in a

**Figure 1** WHO's Surgical Safety Checklist (reproduced with the permission of the WHO; available from http://www.who.int/patientsafety/safesurgery/tools_resources/SSSL_Checklist_finalJun08.pdf).

subset of theatres that covered a diversity of surgical specialties (including general, orthopaedic, oral-maxillofacial and vascular surgery). Interviewees were recruited after a period of observations had been completed, allowing the researchers to familiarise themselves with the setting and the organisation of theatres and theatre teams. Recruitment was guided by purposive sampling so that the sample was diverse in terms of seniority of staff (trainees to consultants and theatre managers), subspecialty and disciplines (see table 1). No one approached declined to be interviewed (although one staff member declined to have

her interview recorded on religious grounds). ELA collected the data in two sites; a second observer (see Acknowledgements) collected data in one of the UK sites. ELA analysed the data from all three sites. ELA and MDW independently reviewed a sample of data and together came to agreement on interpretation.

Data were analysed thematically,[22] supported by Nvivo software and guided but not constrained by sensitising concepts derived from the research questions.[23] A single coding framework was iteratively developed, refined and applied to all three data sets. Data were initially coded into basic themes within three global categories—implementation barriers, facilitators and contextual characteristics. In each category, basic codes were grouped into organising themes, and patterns in different sites were identified, compared and contrasted, paying attention to interrelations between themes. In addition, we analysed data from each site to identify patterns of use, completeness and fidelity of checklist compliance.

**Table 1** Number of individual interviews by profession in African and UK sites

| Role/profession | Mbile (African hospital) | Amfield and Tolgrave (UK hospitals) |
|---|---|---|
| Anaesthetists | 4 | 4 |
| Surgeons | 6 | 3 |
| Managers/administrators | 4 | 4 |
| Nurses and theatre Practitioners | 5 | 9 |
| Total | 19 | 20 |

## RESULTS

We interviewed 39 staff: 19 in Mbile and 20 across the two UK hospitals (see table 1). A total of 112 h of observations were undertaken: 60 h over 2 weeks in Mbile,

28 h over 4 days in Amfield and 24 h over 3 days in Tolgrave.

In all three sites, checklist implementation strategies included staff training, in situ demonstrations and awareness-raising at departmental meetings. In Mbile, training was assisted by two different international organisations, but surgeons were trained separately from other staff. This resulted in some mixed messages being circulated within the operating department.

Though in all three hospitals some staff attributed initial problems with checklist compliance to a lack of awareness or understanding, most staff in the UK and African hospitals could describe broadly how it was *intended* to be used. In all hospitals, many staff saw considerable value in the checklist. The UK staff tended to emphasise prevention of rare but potentially catastrophic errors, such as wrong site surgery, and also suggested that checklist use could improve communication and teamwork with 'no hierarchy' (Amfield, anaesthetist).

> We're trying to prevent what are usually rare errors, rare mistakes, you know, the majority of the things on that checklist are done most of the time without the checklist, but every now and then [...] you forget to check if you're operating on the right leg and not the left leg, and that's rare, but on very rare occasions it then leads to a disaster. [Amfield, Anaesthetist, U021]

> It's not something fluffy and friendly, it is actually functional and it is about all respecting one another [Tolgrave, Anaesthetist, U014]

Mbile staff reported that few other protocols or standardised checks were in place in their hospital; some therefore welcomed the prompting and structure provided by the checklist. In contrast to the UK sites, they emphasised the checklist's value in catching more common mistakes—such as forgetting prophylactic antibiotics—but did not mention a role for the checklist in undermining hierarchies or improving teamwork.

> Whenever we do the checklist we identify that some things have been missed [Mbile, Anaesthetist, A005]

### Compliance

Nurses or theatre practitioners in all three hospitals had primary responsibility for initiating the use of each checklist section, but differences in the way the checks were performed were evident. In the UK hospitals, staff performed checks out loud in front of all team members who were present. In Mbile, checks were undertaken by an individual nurse, who ticked boxes based on his/her perceptions of what had happened or quietly checked on a one-to-one basis whether designated individuals had remembered specific tasks.

> [During the procedure] I see the nurse filling in the checklist, which he then puts on top of the cupboard. He doesn't speak to anyone in the process, except to talk quietly with one of the surgical residents. [Mbile fieldnotes]

Though our study was not designed as a systematic audit, interviews and observations suggested that checklist use, completion and fidelity, while not perfect, were more consistent in the UK settings compared with Mbile. In the UK hospitals, the checklist was *used* in all procedures that we observed, documentation was fully *completed* and interviewees reported internal audits showing checklist use in over 90% of procedures. *Fidelity* was more variable: in some instances, teamwork was undermined by staff being distracted, dismissive or absent during checks. During Time Out, staff often continued with other preparatory tasks, meaning it was not always the 'moment of silence it's supposed to be' (nurse, Tolgrave). Sign In, supposed to be done before induction of anaesthesia, was often performed at the same time as the Time Out when the patient was already anaesthetised. Sign Out was often rushed or cursorily performed, though equipment counts were extremely thorough.

> The scrub nurse and the student Operating Department Practitioner (ODP) are sort of removed, standing over by their metal trolley preparing some of the equipment and talking; they're not really listening to the rest of the group [surgeon, anaesthetist, ODP] who are crowded around patient (who's already anaesthetised), but they answer the questions when they're called to, and they do all the checks. [Amfield fieldnotes]

> At the end of the operation, just as the patient is being moved from the operating table onto a trolley, and they're waiting for him to wake up [..] the anaesthetist says "oh, we haven't done the sign out, oh, we should do the sign out". By this point the surgeon's already left the theatre [Amfield fieldnotes]

In Mbile, *use* of the checklist was highly inconsistent: participants' estimates ranged from 'always' to 'hardly ever', and direct observations induced a Hawthorne effect, where checklist use increased from few procedures early in our observation period to all procedures by the final 2 days. When staff were under pressure due to staff shortages, emergencies or lengths of shifts (up to 36 h) not seen in the UK, the checklist was apt to be abandoned altogether.

> Sometimes it's difficult to use this [checklist], due to staff overload, so sometimes if there is an emergency case, they may not fill it in. Rather than fill it in, they might just get the instruments for those guys because of the urgency of the case [Mbile, Theatre nurse, A011]

*Completeness* was reported and observed to be highly variable in Mbile, with no improvement over the period of observations. *Fidelity* was also problematic in Mbile. Checkboxes were often ticked without the requisite information having been obtained or the tasks to which

they referred undertaken, and the timing of checks was haphazard. Sometimes the nurse ticked the Sign In and Time Out checkboxes when the procedure was already underway, rendering them useless. Team introductions were never performed, yet the box was always ticked. Sometimes the Sign Out boxes were ticked before the procedure was completed. This meant that equipment counts were ticked as complete when the equipment was still in use, and specimens were recorded as correctly labelled before they had been removed from the patient. If equipment counts were performed at all in Mbile, they tended to be unreliable, completed in silence (not out loud as specified by the WHO guidance) by a single nurse at variable points during the procedure using a form that did not list all relevant equipment.

> [During the procedure] I check the checklist: the nurse had (at some point before the end of the operation) filled the entire thing in, including "specimen is correctly labelled"—even though the specimen-to-be was still inside the patient. [Mbile fieldnotes]

> The nurse fills in the back side of the form (the equipment count chart), even though all the equipment used in the operation has been cleared away, and completes the sign out column of the checklist, without talking to anyone. [Mbile fieldnotes]

### Factors influencing checklist implementation

Some reasons for variable compliance were similar between the UK and African sites; others were different, and some problems that appeared to be the same had different underlying mechanisms. A shared barrier to consistent compliance across the UK and African settings was a perception among some that the checklist was futile, or that some checks (particularly the nontechnical ones) were a waste of time.

> The anaesthetist refuses to say his name. He says "I think we all know each other by now [...] it's ridiculous" [Amfield fieldnotes]

> No, no, no [we don't do introductions] We know each other! I know the anaesthetist, I know the nurse. [Mbile, Surgeon, A017]

Perceptions that some elements of the checklist were pointless were partly based on the (demonstrably false) view among some staff at all three hospitals that incidents the checklist was designed to prevent would never happen in their environments. In all hospitals, the persuasive power of sentinel events—even very serious incidents (including, in Mbile, major operations on the wrong patients and a wrong site surgery in Amfield)— was variable.

> I can clearly remember two occasions where the surgeon was just very dismissive and it was like, 'oh I'm not doing this, I don't need to do this, I'm not' you know…and

literally just walked away. [Amfield, Operating Department Practitioner, U020]

> Even though training on the checklist was given for surgeons, they don't use it, they don't believe in this bit of paper, because mostly they said, 'we don't mistake the identity of the patient, it doesn't happen that we get the wrong patient' [Mbile, Theatre nurse, A012]

Differences in factors influencing compliance, and their underlying mechanisms, derived from the very different contexts of implementation. Perhaps the most obvious difference between the settings concerned material resources. In all settings, the technical items were broadly accepted as established good practice. In the UK hospitals, equipment and medicines were available to address these items.

Mbile's infrastructure was very different. The electricity supply, for example, was unreliable; even basic equipment, such as gauze, was not reliably available. Such resource shortages meant that it was not always possible to comply with the technical items on the checklist, yet nurses often ticked the boxes nonetheless. For example, there were no surgical markers at Mbile hospital, but documentary analysis showed the question 'is the site marked?' was always ticked 'yes'. The box for prophylactic antibiotics was also among those most consistently completed ('yes' or 'not applicable' were usually ticked), but, with only a limited range of antibiotics available and no hospital policy regarding administration of prophylactic antibiotics, doctors sometimes delayed antibiotic use in case of subsequent postoperative infections. There were too few pulse oximeters available for the number of operations being performed. Anaesthetists were fully aware of the value of pulse oximeters, but were often unable to access one; absence of the device rarely led to cancellation of an operation.

> We don't have too many types of antibiotics […] if we have given [the patient] prophylactic antibiotics and organisms have already developed some resistance to the antibiotics, how are we going to treat this guy if he develops some type of infection later on? That is the problem […] but we really don't have a single view in this issue, I mean, some people give antibiotics, some people don't give, it's not a standardised policy [Mbile, Surgeon, A019]

> In set-ups like ours, for example if there is no pulse oximetry, [..] if you consider it as a must we're going to lose many people because often we don't have this pulse oximetry. So if you say now I'm not going to anaesthetise a patient because I have no pulse oximeter definitely we will lose many patients that could have been helped if you operate on them without pulse oximeter. [Mbile, Anaesthetist, A002]

The non-technical items on the checklist were far less dependent on material resources, yet they were much more likely to attract criticism and scepticism from the

staff. Some staff in all hospitals (UK and African) objected to the performance of the checklist in its entirety; more often, specific items were the subject of complaint. In the UK sites, a recurring objection was that the checklist duplicated pre-existing systems and checks, including, for example, long-standing equipment counting procedures. Perceptions of 'duplicative' or 'useless' checks contributed to the view of some in all settings that the checklist was an illegitimate bureaucratic intrusion, and resentment towards it intensified during busy periods, emergencies or staff shortages. The most deeply felt and repeatedly vocalised objections came from surgeons. Given the strongly hierarchical team dynamics in both the UK and African hospitals, surgeons' behaviour and attitudes in relation to the checklist were highly influential. This made it difficult for others to directly challenge defiant or dismissive surgeons.

> I think it's difficult because there is a hierarchy within the theatre complex, or within the medical profession full stop, and the theatre staff might feel intimidated by some of the environments they are in. Some of them are forceful enough and have a voice, but others are quite timid and probably quite reluctant to actually speak up. [Tolgrave, Anaesthetist, U012]

> The surgeon may not volunteer to cooperate to fill that checklist. […] The nurse may stop her work, or may do it without filling the checklist. It may depend on the confidence of the nurse. [Mbile, Theatre Nurse, A013]

Despite these objections, use of the checklist was high in the UK hospitals. Staff in these hospitals had been highly sensitised to issues of patient safety through exposure to their institutions' policies, and though they did not see all patient safety interventions as positive, they felt that they had 'a microscope on us about patient safety' (Tolgrave, nurse). Audit and feedback of data on checklist use and relevant outcomes (eg, infection rates) were used to support checklist implementation and were tied to sanctions, which could include disciplinary action against staff who refused to comply.

> You know I do not mind being part of any audit at all because I think it's great, we won't find anything or improve anything unless we do. [Tolgrave, ODP, U001]

> We have to be careful because you know, it's in our registration as well to follow this protocol isn't it, so [even] if the surgeon is not keen to do it, we have to tell them, you have to stop and do it [Tolgrave, ODP, U 008]

These UK audits were not, however, perfect instruments, not least because they generally failed to capture *how* checklists were being used. Beyond formal arrangements, some local leaders acted as champions for the checklist. Surgeons, anaesthetists and senior nurses in both UK hospitals took on this role, and were important in leading by example and supporting junior or non-surgical staff when there was 'push-back'.

> We are there to iron out any problems, or if anybody gives them grief we can go in and fire it back if needed. [Tolgrave, Site coordinator, 002]

In Mbile, introducing the checklist involved much more far-reaching change than in the UK. The absence of an established tradition of equipment counts, for example, meant that the checklist was a much more disruptive and demanding intervention in the African setting. Items that were non-controversial in a UK setting, such as documenting patient consent, had different meaning in a context where patients often lacked literacy and surgery was considered a privilege. The surgical checklist was one of the first explicit patient safety interventions introduced in Mbile. The staff had little prior exposure to the ideas and principles of the science of safety, and systems to support patient safety were poorly developed. The hospital lacked an established culture of audit; there was little routine collection of clinical data and the staff had no clear sense of the impact of the checklist on patient safety. There was no monitoring of checklist use, and no consequences for non-compliance.

> I asked every department [if they want to do an audit] and they have not volunteered […] I think they're busy. It is not familiar, clinical audit. Most health workers don't know what clinical audit means. [Mbile, Administrator, A001]

> We really don't have the real number [for infection rates], how many patients have been affected, because we haven't applied those measurements, we really don't know [Mbile, Surgeon, A020]

> It's ok, nothing happens. For those who don't fill in the checklist there is no problem, no consequence. [Mbile, Theatre nurse, A011]

Team dynamics were especially challenging in Mbile. Though local championing of the checklist was not absent, it was not highly visible either. Surgeons were the only doctors in the room: anaesthetists were not physicians, but held a BSc in anaesthesia. In observations, challenge over checklist use was very rarely witnessed. Wider societal hierarchies and cultural norms, including those relating to age, gender and education, resulted in particularly steep authority gradients. Nurses were often female and younger than the (exclusively) male surgeons, and had been socialised to be deferential and submissive. The working environment was described as lacking in transparency and accountability, particularly in relation to staff dismissals or promotions.

> There were complaints that there is impartial [unfair] treatment between people. This is one of the problems from the managerial aspect. [Mbile, Anaesthetist, A002]

The external sociolegal context in Mbile was one that the staff felt was at best unreliable. At worst it was seen

A patient admitted for cholecystectomy suffered hypoxic brain injury and died following surgery. Subsequently, two staff members (not the surgeon) were threatened with guns by the patient's family, who said the surgical team had killed the patient. The two staff members were later arrested and criminal charges brought against one of them. One of the questions asked during the police investigation was whether a pulse oximeter had been used. It had not: according to staff, no pulse oximeter was available for use, even though the checklist requiring use of this equipment was, officially, in use at the hospital. Adoption of the checklist by the hospital, without adequate provision of the resources necessary for compliance, had left low-status front-line staff more socially and legally vulnerable, as it could be argued that they had failed to comply with hospital policy. Moreover, at all levels of the institution there was a great deal of uncertainty about staff rights and their employers' responsibilities in such circumstances. The accused staff were not provided with any legal representation for some weeks.

**Figure 2** Description of events following an adverse event in Mbile, illustrative of the wider sociolegal context.

as corrupt and unjust, offering little protection to vulnerable, low-status individuals. Interviews suggested that challenging those in authority could therefore feel very risky and have unpredictable results, perhaps damaging career prospects or even worse (figure 2). The checklist thus introduced new, unintended risks for staff.

## DISCUSSION

The WHO's implementation manual identifies two purposes for the surgical checklist: ensuring consistency in patient safety and introducing (or maintaining) a culture that values achieving it.[24] Our study suggests that hospitals in high-income and low-income countries may experience challenges in delivering on these aims, even when many staff (non-physicians in particular) can see value in the checklist. An important finding is the extent to which hospitals in low-income and high-income countries encountered the same obstacles in implementing the checklist; the key differences related to contexts. Material contexts were perhaps most vividly distinct: resource shortages meant that it was often impossible to comply with the checklist's technical requirements in Mbile. Resource and infrastructure constraints in low-income countries have been well described previously,[25–29] suggesting that the problems in our African study hospital were unlikely to be unique. Kwok et al,[9] for example, similarly identified insufficient numbers of pulse oximeters as a barrier to checklist compliance, and suggest that the hospitals in the original pilot study were perhaps not representative of those in low-income settings that lack basic resources. Greater acknowledgement may therefore be needed that additional material resources may well be required to implement the checklist in low-income countries. Many of the non-technical items on the checklist do not, of course, require additional material resources, but rather changes in communication practices and teamwork. Hierarchical team dynamics undermining of safety were evident in all three hospitals in our study, but were more pronounced in the African setting. The assumption that improvements in communication and team interactions automatically follow the introduction of a checklist[4 30] is thus open to question. Our study points to a much more complex interrelation between checklist

procedures, context, 'culture' and behavioural changes, and confirms that checklists should not be regarded as magic bullets.

Contextual differences meant that Mbile had much more to do to implement and secure compliance with the checklist, and it lacked features that supported implementation in the UK sites. Local policies, institutional focus and support, well-established audit and data collection systems, as well as clear lines of accountability with consequences for staff and hospitals for non-compliance were all more pronounced (if not always perfect) in the UK hospitals than their African counterpart. They were important indirectly, by signalling the institutional priority given to patient safety,[18] and directly (eg, disciplinary action). Despite these advantages, the UK sites also showed evidence of persisting problems of fidelity, audits that focused too much on documentation of box-ticking and resentment of perceived top-down initiatives.

Our study produced some disquieting evidence that poor checklist implementation in low-income settings might not only fail to reduce patient safety risks, but also *introduce* new risks for staff and/or patients. Previous work has also shown that, even in well-resourced settings, checklists may not consistently deliver positive effects, and may potentially produce some paradoxical or harmful effects including inhibition of team processes.[5 31] These may be even more consequential in a low-income setting. The style of checks in Mbile limited improvements in team communication, and was directly contrary to the design intent of creating collective awareness. The authoritarian status of the surgeon elite contributed to these unhelpful features of team dynamics, which were exacerbated by wider societal hierarchies and lack of transparency in institutional processes. Though unhelpful team dynamics were also sometimes evident in the UK hospitals, introducing a checklist in a professional environment characterised by a lack of accountability and transparency raised additional ethical concerns in Mbile: it made some staff feel jeopardised legally, professionally and personally, and it encouraged them to make misleading records of what had actually been carried out. Thus, although many staff welcomed

- Checklist implementation will be optimised when part of a broader, multi-faceted cultural and organisational programme to strengthen patient safety
- Introduction of the checklist does not automatically lead to changes in teamwork and communication, despite the inclusion of non-technical items. Hierarchical dynamics within the operating theatre may be particularly prominent in some low-income contexts, exacerbated by the wider socio-legal context
  - Improvements in communication and teamwork require direct interventional focus
  - Whole team training (rather than separate training for different professional groups) is important
- Assumptions about implementation contexts (e.g. resources, policies, systems in place) encoded within the checklist require careful scrutiny
  - Performance of the checklist may not be the only change(s) required
  - In resource-constrained settings especially, existing systems and resources may need to be strengthened for the checklist to be used with completeness and fidelity (e.g. provision of pulse oximetry)
- Local leadership and championing of the checklist is important and needs to:
  - Include support from the highest institutional level
  - Include senior staff, particularly senior surgeons
  - Include leaders from all professional groups
  - Be visible in theatres
- Monitoring (including process and impact measures) must be in place and fed back to all staff
- Poor compliance (use, completeness and/or fidelity) may not only fail to strengthen surgical safety - it may also introduce risks for staff and/or patients in some environments

**Figure 3** Lessons for checklist implementation in high-income and low-income country settings.

attempts to introduce structure and standardisation to the preparation and organisation of procedures, the benefits for patient safety were unclear, especially as no audit data were available to establish outcomes.

None of this is to deny the potential of the checklist to deliver significant benefits for the safety of patients in low-income settings, but it is to emphasise the need for a high level of sensitivity to cultural and economic contexts in the design and implementation of interventions. Introduction of the checklist may not, of itself, address the underlying deficits in the clinical processes the checks cover. Workers cannot comply with checklist items that require material resources (eg, antibiotics and pulse oximeters) that are not available in their organisations. Many non-technical items, of course, represent potentially low-cost ways to improve care and may thus be attractive for resource-constrained environments, particularly when many healthcare interventions are so expensive. But attempts to change the status quo in these settings should be informed by a sound evidence base, attention to unintended consequences and recognition of the influence of local histories.[32] Though the volume of research on patient safety in low-income countries is now increasing,[33] [34] little is known about which kinds of strategies are most effective in addressing patient safety issues in these settings.[35]

Our study suggests that explicit interventional focus on improving team dynamics and communication may be even more critical in African healthcare settings than in high-income countries. Professional groups—in whatever setting—should be trained together on the checklist, not separately as occurred in Mbile. Additional interventional focus on relevant clinical systems may also be necessary. For example, our findings suggest that where equipment count practices are not well established, focused training, agreed procedures and ongoing support to implement these practices are needed if the use of checklists is to be meaningful. Our findings also reinforce lessons from improvement science more broadly concerning the importance of collection and feedback of data; in low-income settings where audit is not such an established feature of institutional governance, additional capacity building may be needed.[36]

The critical role of local leads in motivating the adoption of new ideas[37] means that particular efforts must be made to secure buy-in from surgeons and to ensure that junior or non-medical staff are assured of institutional support. Multiple, senior champions from across the relevant disciplines are needed to ensure that the leadership is visible and persuasive. Another effective tactic may be to empower theatre staff to call on supportive senior surgeons ('champions') for back-up when they

face 'push back' from other staff.[38] While high-level institutional strategies were seen by some as misdirected or problematic in the UK settings, the comparison with Mbile nonetheless underscores the value of all front-line staff having support and clear leadership for their efforts to improve safety. Such institutional support should also include the judicious use of 'hard edges'[38] and mechanisms through which there are predictable consequences for staff—at all levels—when patient safety is jeopardised. As patient safety becomes more firmly established on local and national health agendas in low-income countries, for example, through initiatives such as the WHO's Global Patient Safety Challenge,[39] institutional and infrastructural support for safety is likely to improve as well. Attention to strategies likely to improve implementation (figure 3) may help with this.

Our study has a number of limitations. It did not include data on outcomes. Mbile had only been using the checklist for 1 year, in contrast to 2 years in UK hospitals, so it was at a different stage of implementation. Despite the importance of training to secure checklist compliance,[40] the precise nature of what was provided in the three sites (especially in Mbile) was unclear, due to the unreliable (and in some cases conflicting) accounts of events. A clear observer effect was found in Mbile, and may also have occurred in UK hospitals. Observations over a longer period in all sites might have identified more influences on checklist compliance. Nonetheless, our study has yielded important lessons for implementing surgical checklists in different settings.

Our work demonstrates that assumptions about context are encoded into patient safety tools. Optimising checklist deployment requires careful analysis of these assumptions, and of the extent to which they match local set-ups or require the introduction of new systems and practices. It also demonstrates that changes in teamwork practices do not automatically follow checklist introduction, even where non-technical items are included, but rather require explicit, interventional focus. Checklists are neither a 'quick-fix' nor a tool that can be effectively implemented in isolation; in resource-constrained settings, they are especially unlikely to be free of costs and risks. Safety checklists are most likely to be effective and sustainable when implemented as part of broader, multifaceted programmes addressing social, behavioural, logistical and organisational issues,[14 26 41 42] where there is strong institutional focus on patient safety, multidisciplinary leadership, monitoring systems in place and consequences at all levels for non-compliance.

**Acknowledgements** The authors sincerely thank the staff of all three hospitals for their time and participation in the study. Our thanks to Joel Minion, who collected the data from one of the UK sites. We would also like to thank Dr Wynne Aveling for the clinical expertise he offered during data collection.

**Contributors** ELA and MDW jointly conceived of and designed the study with support from PM. ELA negotiated access, recruited participants and conducted the fieldwork in one UK site and the African site. ELA analysed the data from all three sites. ELA and MDW conducted the literature review, with additional suggestions from PM. ELA and MDW led the writing of the manuscript. PM contributed clinical expertise. All authors contributed to the interpretation of the findings and the drafting of the manuscript and approved the final version. ELA is the guarantor.

**Funding** We thank the Higher Education Innovation Fund Impact Award; Wellcome Trust Senior Investigator Award WT097899MA; and the Department of Health Policy Research Programme (reference number0 770 017) for providing funding for this study.

**Competing interests** None.

**Ethics approval** Ethical approval was granted by the Leicestershire, Northamptonshire and Rutland Research Ethics Committee (number 10/H0406/77), and the Institutional Ethical Review Board for research in the African site.

**Provenance and peer review** Not commissioned; externally peer reviewed..

**Data sharing statement** ELA had full access to all of the data in the study and can take responsibility for the integrity of the data and the accuracy of the data analysis. MDW had full access to the data from the UK sites and the opportunity to access the African data. For ethical reasons, full data were not shared with PM.

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
