## [Reviewer comments · BMJ Open]

Some articles will have been accepted based in part or entirely on reviews undertaken for other BMJ Group journals. These will be reproduced where possible.

ARTICLE DETAILS

TITLE (PROVISIONAL)	A qualitative study comparing experiences of the surgical safety checklist in hospitals in high- and low-income countries
AUTHORS	DixonWoods, Mary; Aveling, Emma-Louise; McCulloch, Peter

VERSION 1 - REVIEW

REVIEWER	Liam Donaldson Chair in Health Policy Institute of Global Health Innovation Imperial College London UK
REVIEW RETURNED	30-Apr-2013

THE STUDY	If title of paper was changed methods would be appropriate. Too sweeping currently. First key message "in the UK" is too big a generalisation from the narrow study base.
RESULTS & CONCLUSIONS	The research question as implied in the title of the paper is very broad in attempting to evaluate barriers to implementation of the surgical checklist in two continents. The study population in contrast is quite narrowly representative. Accepting that a qualitative study should not be expected to provide generalisable findings, more contextual material about experience with checklists in low income countries would be helpful. It would also be helpful to readers who are trying to implement standardised practice in poorly resourced hospitals around the world to be given insight into how to overcome barriers to implementation of measures that could bring improved quality and safety of care.
GENERAL COMMENTS	The title of the paper could be misleading in appearing to compare two continents. There is only one African hospital and 10 anaesthetists/surgeons. The study is very small in scale. Is the statement in the key messages section (page 2, lines 22 - 23): "Consistent use, completion and fidelity of checklist deployment [.....] appears higher in the UK compared with a low-income setting" warranted when so few hospitals were studied? The abstract and the early part of the discussion seem to lean to the negative, yet in the results section it is stated (page 6, line 26): "In all sites, many staff saw considerable value in the checklist." Another example of the emphasis given to findings is (page 10, lines 25 - 27): "The absence of an established tradition of equipment counts, for example, meant that the checklist was a much more disruptive and demanding intervention in the African setting." Rightly, this is stated neutrally as it is in the results section.

	However, it could be picked up in the discussion section to discuss the difficulties of adopting beneficial interventions. Indeed, more generally the authors could talk more fully about the challenges of moving the 'status quo' where there are benefits to safety. In the discussion (page 11, lines 34 - 35), the authors state: "Our finding also challenge the assumption that few additional resources or changes to clinical systems would be needed to implement the checklist." This could seem a little churlish given that many other innovations in healthcare are expensive and disruptive. Overall, the authors should look at the balance of their conclusions in relation to their findings in both the discussion and abstract sections of the paper. Readers of the paper, particularly those from low-income settings might appreciate a more explicit discussion on how to overcome cultural, behavioural and logistic barriers to adoption of the checklist (based on the authors' findings).
--	---

REVIEWER	Dr Ashley Kable Associate Professor Deputy Head of School (Research) School of Nursing and Midwifery University of Newcastle New South Wales Australia No competing interests.
REVIEW RETURNED	14-Jun-2013

THE STUDY	References: There are 2 lists of 33 references. The first list contains references that do not appear to be consistent with the in-text referencing used. Suggest that this should be corrected
GENERAL COMMENTS	This manuscript is well written and provides useful qualitative insights into the issue of achieving compliance with the implementation of the WHO surgical safety checklist in high and low income country contexts. Specific comments and questions: Comments:  1. The data presented in this manuscript provide compelling support for the identified differences in use, completions and contextual aspects of compliance with implementation of the surgical safety checklist. 2. The conclusions are congruent with the findings presented from this study. Questions  1. How was recruitment in this study undertaken? Was there potential for self selection bias in the process of recruitment of participants? 2. I note that the 2 UK hospitals in the study were also participating in a larger research project on culture and behaviour relating to patient safety. Do the authors consider that this may have implications for the transferability of the findings of this study to other UK hospitals? 3. What aspects of rigour and trustworthiness did the researchers apply in the conduct of this study? For example: Was the analysis of data subject to peer/expert review? Was member checking of

	interview transcripts or preliminary analyses conducted? 4. In the first line of the results section there is a reference to Table 2. Should this read as (see Table 1)? 5. References: There are 2 lists of 33 references. The first list contains references that do not appear to be consistent with the in-text referencing used. Suggest that this should be corrected.
--	---

VERSION 1 – AUTHOR RESPONSE

Reviewer: Liam Donaldson
 Chair in Health Policy
 Institute of Global Health Innovation
 Imperial College
 London
 UK

1. If title of paper was changed methods would be appropriate. Too sweeping currently.
2. First key message "in the UK" is too big a generalisation from the narrow study base.
3. The research question as implied in the title of the paper is very broad in attempting to evaluate barriers to implementation of the surgical checklist in two continents. The study population in contrast is quite narrowly representative.

RESPONSE to comments 1-3: We appreciate the reviewer drawing our attention to the need to tighten the language in these sections. We have now amended the title and the first key message to specify that the comparison is between hospitals/case study sites in these settings and avoid misleadingly implying we compared countries or continents.

4. Accepting that a qualitative study should not be expected to provide generalisable findings, more contextual material about experience with checklists in low income countries would be helpful.

RESPONSE: We have now incorporated specific reference to studies of checklist impact in low- and middle-income countries in the Introduction (p. 4) and Discussion (p.11). Like studies in high-income settings, these studies in low-income countries indicate variation in impact and compliance. Little contextual material has been published about experiences of implementation processes that can help explain this variation or how to optimise checklist implementation in diverse settings. Our study aims to contribute to addressing precisely this gap in research about experience with checklist implementation processes in low-income countries, and understanding how/if they differ from high income hospital experiences.

5. It would also be helpful to readers who are trying to implement standardised practice in poorly resourced hospitals around the world to be given insight into how to overcome barriers to implementation of measures that could bring improved quality and safety of care

RESPONSE: To provide more explicit insight into how barriers may be overcome, we have expanded on some of our suggestions in the 4th-6th paragraphs of the Discussion (pp.12-13). We also provide key 'lessons for implementation' in Box 3, and have now changed the mention of this box in the Discussion section to give it more prominence.

6. The title of the paper could be misleading in appearing to compare two continents. There is only one African hospital and 10 anaesthetists/surgeons. The study is very small in scale. Is the statement

in the key messages section (page 2, lines 22 - 23): "Consistent use, completion and fidelity of checklist deployment [.....] appears higher in the UK compared with a low-income setting" warranted when so few hospitals were studied?

RESPONSE: We hope we have addressed this point by amending the title and the key messages (see response to comments 1-3).

7. The abstract and the early part of the discussion seem to lean to the negative, yet in the results section it is stated (page 6, line 26): "In all sites, many staff saw considerable value in the checklist."

RESPONSE: We have added this positive finding to the abstract, and hope it now presents an appropriately balanced summary of the findings.

8. Another example of the emphasis given to findings is (page 10, lines 25 - 27): "The absence of an established tradition of equipment counts, for example, meant that the checklist was a much more disruptive and demanding intervention in the African setting." Rightly, this is stated neutrally as it is in the results section. However, it could be picked up in the discussion section to discuss the difficulties of adopting beneficial interventions. Indeed, more generally the authors could talk more fully about the challenges of moving the 'status quo' where there are benefits to safety.

RESPONSE: We have now added further comment to the Discussion section to acknowledge this useful point (pp12-13).

9. In the discussion (page 11, lines 34 - 35), the authors state: "Our finding also challenge the assumption that few additional resources or changes to clinical systems would be needed to implement the checklist." This could seem a little churlish given that many other innovations in healthcare are expensive and disruptive.

RESPONSE: We have acknowledged this important point more fully in the Discussion, though it is important to note that the importance of material resources was not sufficiently acknowledged in the original pilot study. In revising the paper we are now also able to cite a paper published since our original submission (Kwok et al 2013, Annals of Surgery) which also acknowledges that in resource-poor settings additional resources may be needed, and suggests that the low-income country hospitals included in the original pilot study were not representative of the many hospitals in low-income countries that lack basic resources (see e.g. Lavy et al, 2011; Kotagal et al, 2009).

10. Overall, the authors should look at the balance of their conclusions in relation to their findings in both the discussion and abstract sections of the paper.

RESPONSE: We have amended the abstract (please see response to comment 8). We have also added to the Discussion (p. 11 and p.12) to emphasise that, while implementation was indeed challenging and often sub-optimal, many staff (particularly non-physician staff) saw value in the introduction of the checklist.

11. Readers of the paper, particularly those from low-income settings might appreciate a more explicit discussion on how to overcome cultural, behavioural and logistic barriers to adoption of the checklist (based on the authors' findings).

RESPONSE: We hope that we have addressed this comment through our response to comment 5.

Reviewer: Dr Ashley Kable
Associate Professor
Deputy Head of School (Research)
School of Nursing and Midwifery
University of Newcastle
New South Wales
Australia

No competing interests.

This manuscript is well written and provides useful qualitative insights into the issue of achieving compliance with the implementation of the WHO surgical safety checklist in high and low income country contexts.

Specific comments and questions:

Comments:

1. The data presented in this manuscript provide compelling support for the identified differences in use, completions and contextual aspects of compliance with implementation of the surgical safety checklist.
2. The conclusions are congruent with the findings presented from this study.

RESPONSE: We thank the reviewer for these comments.

Questions

1. How was recruitment in this study undertaken? Was there potential for self selection bias in the process of recruitment of participants?

RESPONSE: We have now elaborated the recruitment process more fully (p. 5). Given that interviews were conducted following a period of observation (allowing familiarisation with the staff and setting), that purposive sampling was used and that no one approached declined to be interviewed, we do not believe that the data were affected by self-selection bias.

2. I note that the 2 UK hospitals in the study were also participating in a larger research project on culture and behaviour relating to patient safety. Do the authors consider that this may have implications for the transferability of the findings of this study to other UK hospitals?

The two UK hospitals were not participating in an interventional study, but in a series of case studies examining culture and behaviour relating to quality and safety in the NHS. While two case studies are unable to “represent” the whole of the NHS, we believe that the findings are likely to be broadly transferable.

3. What aspects of rigour and trustworthiness did the researchers apply in the conduct of this study? For example: Was the analysis of data subject to peer/expert review? Was member checking of interview transcripts or preliminary analyses conducted?

RESPONSE: Formal member checking was not undertaken, though the findings were discussed with two participating sites and staff in these sites accepted them as accurate. The data were analysed systematically, with emerging themes discussed among the research team.

4. In the first line of the results section there is a reference to Table 2. Should this read as (see Table 1)?

RESPONSE: Thank you – this has been corrected in the text

5. References: There are 2 lists of 33 references. The first list contains references that do not appear to be consistent with the in-text referencing used. Suggest that this should be corrected.

RESPONSE: The referencing has now been corrected.

VERSION 2 – REVIEW

REVIEWER	Associate Professor Ashley Kable University of Newcastle Australia No competing interests
REVIEW RETURNED	15-Jul-2013

- The reviewer completed the checklist but made no further comments.